# Multi-Person 3D Motion Prediction with Multi-Range Transformers

**Jiashun Wang**[1]    **Huazhe Xu**[2]    **Medhini Narasimhan**[2]    **Xiaolong Wang**[1]

[1]UC San Diego        [2]UC Berkeley

jiw077@ucsd.edu {huazhe_xu,medhini}@berkeley.edu xiw012@eng.ucsd.edu

## Abstract

We propose a novel framework for multi-person 3D motion trajectory prediction. Our key observation is that a human's action and behaviors may highly depend on the other persons around. Thus, instead of predicting each human pose trajectory in isolation, we introduce a *Multi-Range Transformers* model which contains of a local-range encoder for individual motion and a global-range encoder for social interactions. The Transformer decoder then performs prediction for each person by taking a corresponding pose as a query which attends to both local and global-range encoder features. Our model not only outperforms state-of-the-art methods on long-term 3D motion prediction, but also generates diverse social interactions. More interestingly, our model can even predict 15-person motion simultaneously by automatically dividing the persons into different interaction groups. Project page with code is available at https://jiashunwang.github.io/MRT/.

## 1 Introduction

Given a few time steps of human motion, we are able to forecast how the person will continue to move and imagine the complex dynamics of their motion in the future. The ability to perform such predictions allows us to react and plan our own behaviors. Similarly, a predictive model for human motion is an essential component for many real world computer vision applications such as surveillance systems, and collision avoidance for robotics and autonomous vehicles. The research on 3D human motion prediction has caught a lot of attention in recent years [44, 43], where deep models are designed to take a few steps of 3D motion trajectory as inputs and predict a long-term future 3D motion trajectory as the outputs.

While encouraging results have been shown in previous work, most of the research focus on single human 3D motion prediction. Our key observation is that, how a human acts and behaves may highly depend on the people around. Especially during interactions with multiple agents, an agent will need to predict the other agents' intentions, and then respond accordingly [54]. Thus instead of predicting each human motion in isolation, we propose to build a model to predict multi-person 3D motion and interactions. Such a model needs the following properties: (i) understand each agent's own motion in previous time steps to obtain smooth and natural future motion; (ii) within a crowd of agents, understand which agents are interacting with each other and learn to predict based on the social interactions; (iii) the time scale for prediction needs to be long-term.

In this paper, we introduce Multi-Range Transformers for multi-person 3D motion trajectory prediction. The Transformer [63] has shown to be very effective in modeling long-term relations in language modeling [16] and recently in visual recognition [17]. Inspired by these encouraging results, we propose to explore Transformer models for predicting long-term human motion (3 seconds into the future). Our Multi-Range Transformers contain a local-range Transformer encoder for each individual person trajectory, a global-range Transformer encoder for modeling social interactions, and a Transformer decoder for predicting each person's future motion trajectory in 3D.

35th Conference on Neural Information Processing Systems (NeurIPS 2021).

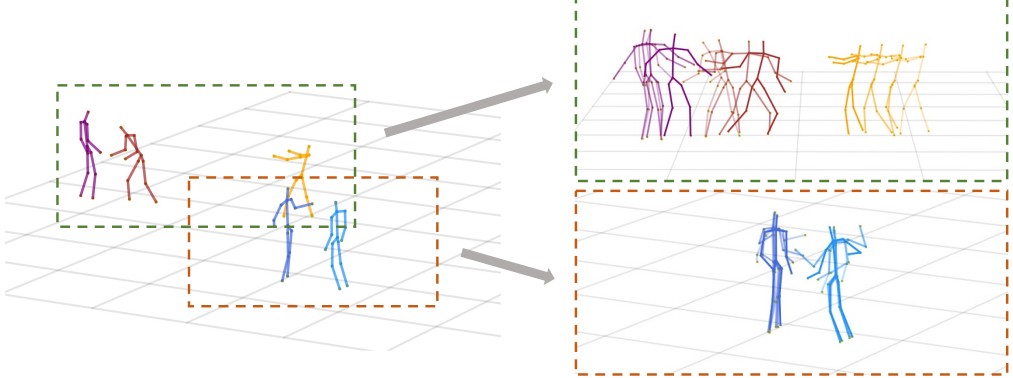

Figure 1: Our motion prediction results. **Left**: The last time step of the input sequence. **Right**: The predicted diverse and continuous motion with multi-person social interactions. We use different color to indicate different persons and the darker color for the further in future for predictions.

Specifically, given the human pose joints (with 3D locations in the world coordinate) in 1-second time steps as inputs, the local-range Transformer encoder processes each person's trajectory separately and focuses on the local motion for smooth and natural prediction. The global-range Transformer encoder performs self-attention on 3D pose joints across different persons and different time steps, and it automatically learns which persons that one person should be attending to model their social interactions. Our Transformer decoder will then take a single human 3D pose in *one time step* as the query input and encoder features as the key and value inputs to compute attention for prediction. We perform prediction for different persons by using different query pose inputs. By using only one time step person pose as the query for the decoder instead of a sequence of motion steps, we create a bottleneck to force the Transformer to exploit the relations between different time steps and persons in the encoders, instead of just repeating the existing motion alone [43].

We perform our experiments on multiple datasets including CMU-Mocap [1], MuPoTS-3D [48], 3DPW [64] for multi-person motion prediction in 3D (with $2 \sim 3$ persons). Our method achieves a significant improvement over state-of-the-art approaches for long-term predictions and the gain enlarges as we increase the future prediction time steps from 1 second to 3 seconds. Qualitatively, we visualize that our method can predict interesting behaviors and interactions between different persons while previous approaches will repeat the same poses as it goes to further steps in the future. More interestingly, we extend the task to perform prediction with $9 \sim 15$ persons by mixing the CMU-Mocap [1] and the Panoptic [24] datasets. We visualize part of the prediction results with multi-person interactions in Fig. 1. We show that Multi-Range Transformers can not only perform predictions with a crowd of persons, but also learn to automatically group the persons into different social interaction clusters using the attention mechanism.

## 2  Related Work

**3D Motion Prediction.** Predicting future human pose in 3D has been widely studied with Recurrent Neural Networks (RNNs) [15, 19, 31, 41, 46, 51, 20, 22, 26, 72, 21]. For example, Fragkiadaki *et al.* [19] propose a Encoder-Recurrent-Decoder (ERD) model which incorporates nonlinear encoder and decoder networks before and after recurrent layers. Besides using RNNs, temporal convolution networks have also show promising results on modeling long-term motion [37, 28, 10, 10, 44, 43, 5]. For example, Li *et al.* [37] use a convolutional long-term encoder to encode the given history motion into hidden variable and then use a decoder to predict the future sequence. While these approaches show encouraging results, most of these studies fix the pose center and ignore the global body trajectory. Instead of solving two problems separately, recent works start looking into jointly predict human pose and the trajectory in the world coordinate [70, 73, 71, 65, 11]. For example, Cao *et al.* [11] introduce to predict human motion under the constraint of 3D scene context. Our work also predicts the human motion with both 3D poses and the trajectory movements at the same time. Going beyond single human prediction, we predict multi-human motion and interaction.

**Social interaction with multiple persons.** Multi-person trajectory prediction has been a long standing problem in decades [25, 47, 67, 52, 75, 7, 6, 23, 18, 49, 36, 42, 56, 8, 34, 61, 69, 62].

For example, Alahi *et al.* [7] present a LSTM [27] model which jointly reasons across multiple individuals in a scene. Gupta *et al.* [23] propose to predict socially plausible futures with Generative Adversarial Networks. However, most of these approaches focus on the global movement of the humans. To model more fine-grained human-human interactions, recent research have proposed to predict multi-person poses and trajectories at the same time [58, 57, 3, 2]. For example, Shu *et al.* [58] propose a MCMC based algorithm automatically discovers semantically meaningful interactive social affordance from RGB-D videos. Adeli *et al.* [2] introduce to combine context constraints in multi-person motion prediction. Inspired by these works, we propose a novel Multi-Range Transformers model which scales up the long-term prediction with even more than 10 persons.

**Transformers.** Transformer is first introduced by Vaswani *et al.* [63] and has been widely applied in language processing [16, 55, 9] and computer vision [66, 29, 50, 14, 35, 17, 12, 13, 60, 68, 59, 45, 39, 53]. For example, Dosovitskiy *et al.* [17] introduce that using Transformer alone can lead to competitive performance in image classification. Beyond recognition tasks, Transformers have also been applied in modeling the human motion [11, 40, 38, 5, 43]. Aksan *et al.* [5] use space-time self-attention mechanism with skeleton joints to synthesize long-term motion. Mao *et al.* [43] propose to extract motion attention to capture the similarity between the current motion and the historical motion. Beyond modeling single human motion, we introduce multi-range encoders to combine local motion trajectory and global interactions, and a decoder to predict multi-person motion.

## 3 Method

### 3.1 Representation

Given a scene with $N$ persons and their corresponding history motion, our goal is to predict their future 3D motion. Specifically, given $X_{1:k}^n = [x_1^n, ..., x_k^n]$ representing the history motion of person $n$ where $n = 1, ..., N$, and $k$ is the time step. We aim to predict the future motion $X_{k+1:T}^n$ where T represents the end of the sequence. We use a vector $x_k^n \in \mathbb{R}^{3J}$ containing the Cartesian coordinates of the $J$ skeleton joints to represent the pose of the person $n$ at time step $k$. In contrast to most previous motion prediction works which center the pose (joint positions) at the origin, we instead use the absolute joint positions in the world coordinate. In our method, $x_k^n$ contains both the trajectory and the pose information. For simplicity, we omit subscript $n$ when $n$ only represents an arbitrary person, e.g., taking $x_{1:k}^n$ as $x_{1:k}$.

### 3.2 Network Architecture

We propose a Multi-Range Transformers architecture, which allows each person to query other persons' and their own history of motion for generating socially and physically plausible future motion in 3D. The network architecture is shown in Fig. 2. The proposed architecture is composed of a motion predictor $\mathcal{P}$ and a motion discriminator $\mathcal{D}$. In the predictor $\mathcal{P}$, two Transformer-based encoders encode the individual (local) and global motion separately and one Transformer-based decoder decodes a smooth and natural motion sequence.

To ensure the smoothness of the motion, the model requires dense sampling of the input sequence. We apply our local-range encoder to each individual's motion to achieve this. For modeling the interaction of all the persons in the whole scene, we apply our global-range encoder to their motions, which performs a relatively sparse sampling of the sequences. And this only need to be calculated once for the whole scene.

The motion discriminator $\mathcal{D}$ is a Transformer-based classifier to determine whether the generated motion is natural. We apply Discrete Cosine Transform (DCT) [4, 44] to encode the inputs for the encoders and the Inverse Discrete Cosine Transform (IDCT) for the decoder outputs. We introduce the architectures for each component as following.

#### 3.2.1 Local-range Transformer Encoder

When predicting one person's motion, we first use our Local-range Transformer encoder to process this person's history motion. We use offset $\Delta x_i = x_{i+1} - x_i$ between two time steps to represent the motion. We apply DCT and a linear layer to $\Delta x_{1:k}$ and then add the sinusoidal positional embedding [63] to get the local motion embedding for $k$ time steps $l_{1:k} = [l_1, ..., l_k]$. We concatenate

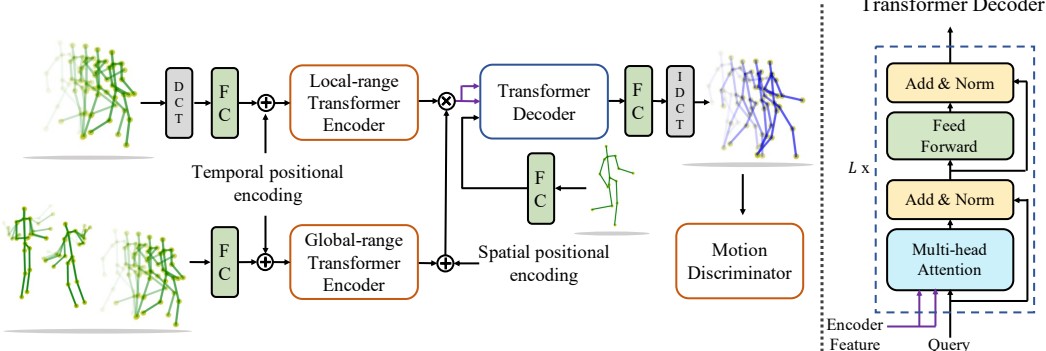

Figure 2: **Network architecture.** Individual input motion is sent to the Local-range Transformer Encoder and all the person's motions are sent to the Global-range Transformer Encoder. $\otimes$ represents concatenate and $\oplus$ represents add. The encoded motion features are used as the key and value together with the query person skeleton for the Transformer Decoder. The output is the future motion prediction results. On the right, we show the architecture of the Transformer decoder. The encoder architecture is similar with the decoder except that the query, key and value are from the same input.

them as a set of tokens $E_{loc} = [l_1, ..., l_k]^T$ and feed them to the Transformer encoder. There are $L$ stack alternating layers in the local-range encoder and we introduce the technique we use in each layer.

Firstly, a Multi-Head Attention is used for extracting the motion information,

$$\text{MultiHead}(Q, K, V) = [\text{head}_1; ...; \text{head}_h]W^O$$
$$\text{where head}_i = \text{softmax}(\frac{Q^i(K^i)^T}{\sqrt{d_K}})V^i \qquad (1)$$

$W^O$ is a projection parameter matrix, $d_K$ is the dimension of the key and $h$ is the number of the heads we use. We use self-attention and get the query $Q_{loc}$, key $K_{loc}$, and value $V_{loc}$ from $E_{loc}$ for each head$_i$,

$$Q_{loc}^i = E_{loc}W_{loc}^{(Q,i)}, \ K_{loc}^i = E_{loc}W_{loc}^{(K,i)}, \ V_{loc}^i = E_{loc}W_{loc}^{(V,i)} \qquad (2)$$

where $W_{loc}^{(Q,i)}, W_{loc}^{(K,i)}, W_{loc}^{(V,i)}$ are projection parameter matrices. $loc$ represents the local-range. We then employ a residual connection and the layer normalization techniques to our architecture. We further apply a feed forward layer, again followed by a residual connection and a layer normalization following [63]. The whole process forms one layer of local-range Transformer encoder. We stack $L$ such Transformer encoders to update the local motion embedding and obtain the local motion feature $e_{1:k} = [e_1, ..., e_k]$ as the output, with $e_i$ represents the feature for time step $i$.

### 3.2.2 Global-range Transformer Encoder

In the global-range Transformer encoder, we aim to encode all the $N$ people's motion in the scene. In our method, this only needs to be calculated one time and then can be concatenated with any person's local motion feature to predict the correspond person's future motion. We first apply a linear layer to each person's motion $x_{1:k}^n$ and plus the sinusoidal positional embedding to get the global motion embedding $g_{1:k}^{1:N} = [g_1^1, ..., g_k^1, ..., g_1^N, ..., g_k^N]$ for $N$ persons in $k$ time steps. We use $L$ layers of Transformers to encode the global motion embedding. We apply the Multi-head Attention mechanism similar to Eq. 1 to the global embedding which calculated as,

$$Q_{glob}^i = E_{glob}W_{glob}^{(Q,i)}, \ K_{glob}^i = E_{glob}W_{glob}^{(K,i)}, \ V_{glob}^i = E_{glob}W_{glob}^{(V,i)} \qquad (3)$$

where $E_{glob} = [g_1^1, ..., g_k^1, ..., g_1^N, ..., g_k^N]^T$ and $W_{glob}^{(Q,i)}, W_{glob}^{(K,i)}$ and $W_{glob}^{(V,i)}$ are projection parameter matrices. $glob$ represents the global-range. Then we feed them to the normalization and feed-forward layers same as local-range Transformer encoder. After applying such $L$ Transformer encoders to the global embedding, we can get the output $o_{1:k}^{1:N}$. We add our spatial positional encoding to the output and get the global motion feature $f_{1:k}^{1:N}$. We concatenate the local $e_{1:k}$ and global $f_{1:k}^{1:N}$ features together as $H = [e_1, ..., e_k, f_1^1, ..., f_k^1, ..., f_1^N, ..., f_k^N]^T$ and feed them to the decoder.

### 3.2.3 Positional Encoding

**Temporal positional encoding.** We apply the sinusoidal positional encoding [63] to the inputs in both local and global-range encoder. This technique is routine in many Transformer-based methods for injecting information about the relative or absolute temporal position to the models.

**Spatial positional encoding.** We propose a spatial positional encoding (SPE) technique on the outputs of the global-range Transformer encoder. Before forwarding to the Transformer decoder, we want to provide the spatial distance between the query token $x_k$ and the tokens of every time step of each person $x_{1:k}^{1:N}$. Intuitively, the location information helps clustering different persons in different social interactions groups, especially in a scene with a crowd of persons. We calculate SPE as,

$$\text{SPE}(x_t^n, x_k) = \exp(-\frac{1}{3J}||x_t^n - x_k||_2^2) \tag{4}$$

where $n = 1, ..., N$, $t = 1, ..., k$, and $x_t^n, x_k \in \mathbb{R}^{3J}$. SPE$(\cdot)$ will explicitly calculate the spatial distance between two persons. All the SPE $(x_t^n, x_k)$ will then add to $o_t^n$ (the output of the global-range encoder) respectively in order to get the global feature $f_{1:k}^{1:N}$.

### 3.2.4 Transformer Decoder

We send the local-global motion feature $H$ together with a static human pose $x_k$ at time step $k$ into the decoder. We use the similar Multi-head attention mechanism as the Transformer encoders. But differently, we take the single pose as the query and use the feature from the encoders to get keys and values. We also apply $L$ layers of Transformer decoder. Specifically, we apply a linear layer to $x_k$ and then get $q$ in order to get the query $Q_{dec}$, we get the key $K_{dec}$ and value $V_{dec}$ both from the local-global motion feature $H$ as,

$$Q_{dec}^i = q^T W_{dec}^{(Q,i)}, \ K_{dec}^i = H W_{dec}^{(K,i)}, \ V_{dec}^i = H W_{dec}^{(V,i)} \tag{5}$$

where $W_{dec}^{(Q,i)}, W_{dec}^{(K,i)}$ and $W_{dec}^{(V,i)}$ are projection parameter matrices. At the end of the decoder, we apply two fully connected layers followed by Inverse Discrete Cosine Transform (IDCT) [4, 44] and output an offset motion sequence $[\Delta\hat{x}_k, ..., \Delta\hat{x}_{T-1}]$ which can easily lead to the future 3D motion trajectory $\hat{x}_{k+1:T}$. The model outputs a sequence of motion directly instead of a pose each time and this design can prevent generating freezing motion [40]. Note we also add residual connections and layer normalization between layers.

### 3.2.5 Motion Discriminator

The design of such encoder-decoder architecture helps to predict the future motion. To ensure a natural and continuous long-term motion, we use a discriminator $\mathcal{D}$ to adversarially train the Predictor $\mathcal{P}$. The output motion $\hat{x}_{k+1:T}$ is given as input to the Transformer encoder with the same architecture of the local-range encoder and we further use another two fully connected layers to predict values $\in \{1, 0\}$ representing that $\hat{x}_{k+1:T}$ are real or fake poses. We use the ground-truth future poses to provide as the positive examples. We train the predictor $\mathcal{P}$ and discriminator $\mathcal{D}$ jointly.

### 3.3 Training and Inference

We train our predictor $\mathcal{P}$ with both the reconstruction loss and the adversarial loss,

$$L_{\mathcal{P}} = \lambda_{rec} L_{rec} + \lambda_{adv} L_{adv} \tag{6}$$

where $\lambda_{adv}$ and $\lambda_{rec}$ are constant coefficients to balance the training loss. We calculate the $L_{rec}$ and $L_{adv}$ as follows,

$$L_{rec} = \frac{1}{T-k} \Sigma_{t=k}^{T-1} ||\Delta\hat{x}_t - \Delta x_t||_2^2$$
$$L_{adv} = \frac{1}{T-k} ||\mathcal{D}(\hat{x}_{k+1:T}) - \mathbf{1}||_2^2 \tag{7}$$

We train our discriminator $\mathcal{D}$ following [33] with loss $L_{\mathcal{D}}$,

$$L_{\mathcal{D}} = \frac{1}{T-k} ||\mathcal{D}(\hat{x}_{k+1:T})||_2^2 + \frac{1}{T-k} ||\mathcal{D}(y_{k+1:T}) - \mathbf{1}||_2^2 \tag{8}$$

where $\hat{x}_{k+1:T}$ is from the predicted motion and $y_{k+1:T}$ is from the real motion. We train the discriminator that classifies the real ones as $\mathbf{1}$, where $\mathbf{1} \in \mathbb{R}^{T-k}$ represents all the poses are natural.

We propose an efficient strategy which is progressively increasing the input sequence length during training and inference. Since Transformer-based encoders are used, there is no limit to the length of the input sequence. During **training**, we provide $x_{1:k}$ to predict $\hat{x}_{k+1:2k}$ ($x_k$ as the query), and we provide $x_{1:2k}$ to predict $\hat{x}_{2k+1:3k}$ ($_{2k}$ as the query) and so on. We sample different lengths of sequence in this way input during training. During **inference**, we predict the future motion in an auto-regressive way. Given $x_{1:k}$, our model predicts $\hat{x}_{k+1:2k}$. Then, given the input and our prediction results, we use them as inputs for our model again to predict $\hat{x}_{2k+1:3k}$, and so on. The advantage of such design is that when predicting longer motions, we still maintain the early motions as inputs to the model, instead of using a fixed length to predict each of the future time steps [44, 43], which may cause the loss of early interactive information. Through the experiment, we find this strategy could largely reduce the error accumulation.

## 4 Experiments

### 4.1 Datasets

We perform our experiments on multiple datasets. CMU-Mocap [1] contains high quality motion sequences of $1 \sim 2$ persons in each scene. Panoptic [24], MuPoTS-3D [48] and 3DPW [64] datasets are collected using cameras with pose estimation and optimization. Consequently, the estimated joint positions in the latter datasets contain more noise and are less smooth in comparison to CMU-Mocap. For multi-person data, there are about $3 \sim 7$ persons in each scene in Panoptic, $2 \sim 3$ persons in MuPoTS-3D each scene and 2 persons in 3DPW. It is worth noting that in the Panoptic dataset, most of the scenes are people standing and chatting, with small movement. The skeleton joint positions in 3DPW are obtained from a moving camera so that there is more unnatural foot skating.

**Dataset settings.** We design two different settings to evaluate our method. The first setting consists of a small number of people ($2 \sim 3$). We use CMU-Mocap as our training data. CMU-Mocap contains a large number of scenes with a single person moving and a small number of scenes with two persons interacting and moving. We sample from these two parts and mix them together as our training data. We make all the CMU-Mocap data consists of 3 persons in each scene. We sample test set from CMU-Mocap in a similar way. We evaluate the *generalization* ability of our model by testing on MuPoTS-3D and the 3DPW dataset with the model trained on the entire CMU-Mocap dataset.

The second setting consists of scenes with more people. For the training data, we sample motions from CMU-Mocap and Panoptic and then mix them. We integrate CMU-Mocap and Panoptic scenes to build a single scene with more people. For the test data, we sample one version from both CMU-Mocap and Panoptic, namely Mix1. And we sample one version from CMU-Mocap, MuPoTS-3D and 3DPW, namely Mix2. There are $9 \sim 15$ persons in each scene in Mix1 and 11 persons in Mix2. The positive motion data for discriminator is sampled from CMU-Mocap's single person motion.

### 4.2 Implementation Details

In our experiments, we give 1 second history motion ($k = 15$ time steps) as input and recursively predict the future 3 seconds (45 time steps) as Sec. 3.3 described. We use $L = 3$ alternating layers with 8 heads in each Transformer. We use Adam [32] as the optimizer for our networks. During training, we set $3 \times 10^{-4}$ as the learning rate for predictor $\mathcal{P}$ and $5 \times 10^{-4}$ as the learning rate for discriminator $\mathcal{D}$. We set $\lambda_{rec} = 1$ and $\lambda_{adv} = 5 \times 10^{-4}$. For experiments with $2 \sim 3$ persons, we set a batch size of 32 and for scene with more people, we set a batch size of 8.

### 4.3 Metrics and Methods for Comparisons

**Metrics.** We first use Mean Per Joint Position Error (MPJPE) [30] without aligning as the metric to compare the multi-person motion prediction results in 1, 2 and 3 seconds. Without aligning, MPJPE will reflect the error caused by both trajectory and pose. We also compare the root error and pose error (MPJPE with aligning) separately. The root error is the L2 root joint position error. Further, we compare the distribution of the movement between the start and end of the outputs on different datasets. Specifically, we measure this movement by calculating the mean L2 distance between the

| method | CMU-Mocap (3 persons) | | | MuPoTS-3D (2 ∼ 3 persons) | | | 3DPW (2 persons) | | | Mix1 (9 ∼ 15 persons) | | | Mix2 (11 persons) | | |
|---|---|---|---|---|---|---|---|---|---|---|---|---|---|---|---|
| | 1 s | 2s | 3s | 1 s | 2s | 3s | 1 s | 2s | 3s | 1 s | 2s | 3s | 1 s | 2s | 3s |
| LTD [44] | 1.37 | 2.19 | 3.26 | 1.19 | 1.81 | 2.34 | 4.67 | 7.10 | 8.71 | 2.10 | 3.19 | 4.15 | 1.72 | 2.58 | 3.45 |
| HRI [43] | 1.49 | 2.60 | 3.07 | 0.94 | 1.68 | 2.29 | 4.07 | 6.32 | 8.01 | 1.80 | 3.14 | 4.21 | 1.60 | 2.71 | 3.67 |
| SocialPool [2] | 1.15 | 2.71 | 3.90 | 0.92 | 1.67 | 2.51 | 4.17 | 7.17 | 9.27 | 1.85 | 3.39 | 4.84 | 1.72 | 3.06 | 4.26 |
| Ours w/o Local | 1.42 | 2.20 | 2.99 | 1.28 | 2.10 | 2.78 | 3.96 | **5.94** | **7.75** | 2.09 | 3.34 | 4.34 | 1.85 | 2.92 | 3.83 |
| Ours w/o Global | 0.99 | 1.71 | 2.50 | 0.92 | 1.67 | 2.50 | 4.17 | 6.85 | 8.91 | 1.77 | 3.10 | 4.19 | 1.42 | 2.29 | 3.06 |
| Ours w/o $\mathcal{D}$ | 1.13 | 1.84 | 2.57 | 0.92 | 1.62 | 2.26 | 4.17 | 6.41 | 8.09 | 1.75 | 3.00 | 4.00 | 1.34 | 2.19 | 2.95 |
| Ours w/o SPE | 1.05 | 1.68 | 2.37 | 0.92 | **1.51** | 2.23 | 3.92 | 6.18 | 7.79 | 1.75 | 3.09 | 4.13 | 1.31 | 2.15 | 2.92 |
| Ours | **0.96** | **1.57** | **2.18** | **0.89** | 1.59 | **2.22** | **3.87** | 6.12 | 7.83 | **1.73** | **2.99** | **3.97** | **1.29** | **2.09** | **2.82** |

Table 1: MPJPE on different datasets. We compare our method with the previous SOTA methods and ablative baselines on predicting 1, 2 and 3 seconds motion. Best results are shown in boldface.

| method | CMU-Mocap Root | | | CMU-Mocap Pose | | | MuPoTS-3D Root | | | MuPoTS-3D Pose | | | 3DPW Root | | | 3DPW Pose | | |
|---|---|---|---|---|---|---|---|---|---|---|---|---|---|---|---|---|---|---|
| | 1s | 2s | 3s | 1s | 2s | 3s | 1s | 2s | 3s | 1s | 2s | 3s | 1s | 2s | 3s | 1s | 2s | 3s |
| LTD [44] | 0.97 | 1.73 | 2.62 | 0.98 | 1.21 | 1.37 | 0.89 | 1.39 | 1.91 | 0.88 | 1.14 | 1.31 | 4.28 | 6.79 | 8.41 | 1.54 | 1.76 | 1.98 |
| HRI [43] | 0.96 | 2.06 | 3.11 | 1.05 | 1.37 | 1.58 | **0.66** | 1.30 | 2.16 | 0.73 | 1.07 | 1.30 | 3.67 | 6.42 | 8.64 | **1.43** | **1.75** | 1.94 |
| SocialPool [2] | 0.96 | 2.01 | 2.96 | 1.03 | 1.41 | 1.71 | 0.96 | 1.38 | 2.21 | 0.72 | 1.08 | 1.30 | 3.76 | 6.86 | 9.07 | 1.60 | 1.95 | 2.15 |
| Ours | **0.60** | **1.12** | **1.71** | **0.79** | **1.05** | **1.22** | 0.67 | **1.25** | **1.86** | **0.69** | **0.99** | **1.19** | **3.42** | **5.69** | **7.30** | 1.52 | **1.75** | **1.93** |

Table 2: Root and pose error on different datasets. We compare our method with the previous SOTA methods on predicting 1, 2 and 3 seconds motion. Best results are shown in boldface.

start and end skeleton joint positions. This surrogate metric is designed to measure whether a method could predict a plausible movement, instead of standing and repeating. We also perform a user study using Amazon Mechanical Turk (AMT) to compare the realness of the predicted motion. We let the user to score each motion prediction from 1 to 5, with a higher score implying the video looks more natural inspired by [65, 74].

**Methods for comparisons.** We argue that for multi-person motion prediction, joint positions are more relevant compared to angle-based representations, since most of the data suitable for this task is obtained by cameras and pose estimation, containing only the position representations. Another consideration for comparison method selection is many related studies only model the pose fixing at the origin, while we need to select methods that could predict absolute motion. We select two competitive state-of-the-art person motion prediction methods: LTD [44] is a graph-based method and HRI [43] is an attention-based method. Both of them allow absolute coordinate inputs, which fits our task and settings well. Most relevant to our work is SocialPool [2], a method which uses GRU [15] to model the motion sequences and proposes to use a social pool to model the interaction. We remove the image input of it as well as the image feature, keeping the sequence to sequence and social pool structures. We use the same data to train these methods.

## 4.4 Evaluation Results

### 4.4.1 Quantitative Results

We report the MPJPE in 0.1 meters of 1 second, 2 seconds and 3 seconds predicted motion on CMU-Mocap, MuPoTS-3D and 3DPW, Mix1 and Mix2 respectively in Tab. 1. In both cases with a small number and a large number of people, our method achieves state-of-the-art performance for different prediction time lengths. We achieve up to 20% improvement when compared to the previous single-person-based methods [44, 43] and achieve up to 30% improvement compared to the multi-person-based method [2]. In SocialPool [2], the same global feature is added to all the persons which interferes with the model's prediction for each individual, especially when there are a large number of people. Because of this, SocialPool's performance is even not as good as the previous single-person-based method. However, in our design the model can use the features corresponding to one person to query the global motion feature which automatically allows it to use the motion information belonging to other persons. Therefore, our method can achieve good results on scenes with any number of people. We also compare the root error and pose error respectively in Tab. 2. The results are in 0.1 meters. Generally, our method can predict the trajectory of the root and the

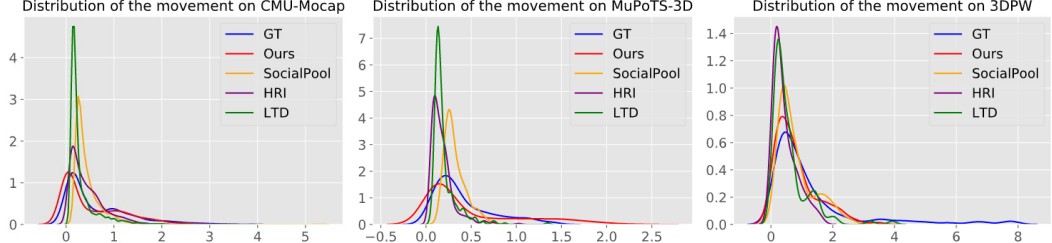

Figure 3: Distribution of the movement between the start and end of the predictions on different datasets. X-axis shows the moving distance. Ours is the most similar to the ground truth while the others intend to predict smaller movement. The model is only trained on CMU-Mocap.

| method | CMU-Mocap | MuPoTS-3D | 3DPW | Mix1 | Mix2 |
|---|---|---|---|---|---|
| LTD [44] | 3.61±0.83 | 3.66±**0.93** | 3.65±0.76 | 3.71±0.93 | 3.75±0.90 |
| HRI [43] | 3.36±0.96 | 3.59±1.25 | 3.76± **0.72** | 3.67±0.89 | 3.71±0.90 |
| SocialPool [2] | 3.49±0.87 | 3.66±1.19 | 3.66±0.86 | 3.62±0.92 | 3.49±1.02 |
| Ours | **3.62**±**0.78** | **3.68**±0.98 | **3.78**±0.82 | **3.74**±**0.83** | **3.77**±**0.82** |
| GT | 3.78±0.76 | 3.85±0.96 | 3.77±0.81 | 3.77±0.87 | 3.88±0.79 |

Table 3: **User study.** We perform a user study using the AMT. We provide the average of the human evaluated score w.r.t. the average ± the standard deviations. Best results are shown in boldface.

poses more accurately, especially for the root prediction. We will also introduce in the following experiments that previous methods often intend to predict a shorter movement.

We show the distribution of the moving distance between the start and the end of the predictions in Fig. 3. It shows that other methods intend to predict a motion with less movement, with much higher value in y-axis close to 0 in x-axis. We observe the results of other methods sometimes just stay in the same spatial location while the ground truth is moving with a large distance. On the contrary, the distribution of our result is very similar to the ground truth on CMU-Mocap and even on completely unseen MuPoTS-3D and 3DPW datasets. The reason is that our Transformer encoders can take any time length motion as input for modeling the long-term motion. And using the decoder to output a $\Delta x$ motion is also beneficial to model the position movement. For user study, we report the average and the standard error of the score in Tab. 3. A higher average score means users think that the results are more "natural looking" and a lower standard error indicates that the scores are more stable. Our results get better reviews consistently across all datasets.

### 4.4.2 Qualitative Results

We provide qualitative comparisons in Fig. 4. Our predictions are more natural and smooth while being close to the real record. It can be seen that RNN-based method (SocialPool [2]) will quickly produce freezing motion, which is consistent with the claims in [5, 40]. Another finding is that when predicting the absolute skeleton joint positions, decoding based on an input seed sequence (HRI [43]) or adding the input sequential residual (LTD [44]) to the output, will make the predicted motion have hysteresis and repeat the history. For example, in a forward motion, the prediction may jump back into temporally unreasonable position and then continue to move forward. We argue the positions of the past sequences have a negative impact on the model's prediction using the previously proposed design. This is also consistent with the conclusion we get in Fig. 3. However, our method, using a static pose as query and predicting a $\Delta x$ sequence, could solve this problem effectively.

### 4.4.3 Ablation Study

We perform ablation study on different modules of our network. We provide the MPJPE results on different datasets in Tab. 1. We remove the local-range encoder, global-range encoder, discriminator and the spatial positional encoding respectively. After removing each module, the overall performance of the model has declined.

| | 1f | 2f | 4f | half | all |
|---|---|---|---|---|---|
| 1 second | **0.96** | **0.96** | **0.96** | 1,03 | 1.03 |
| 2 seconds | **1.57** | 1.58 | **1.57** | 1.71 | 1.73 |
| 3 seconds | **2.18** | 2.22 | 2.26 | 2.44 | 2.50 |

Table 4: MPJPE with different decoder query input length on CMU-Mocap. "f" denotes frame. "half" denotes half of the sequence and "all" denotes the whole sequence.

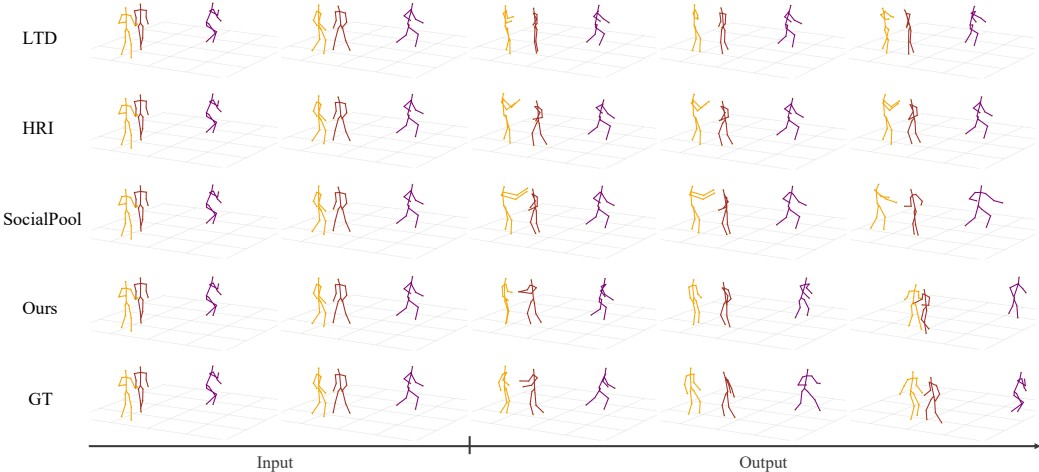

Figure 4: Qualitative comparison with other methods. Left two columns are input and right three columns are outputs. Our result is the closest to the real record and the others fail to predict a walking motion and predict a less accurate interaction motion. The input data is from CMU-Mocap.

We find local-range encoder could largely reduce the prediction error. We also prove the effectiveness of the global-range encoder and an adversial training would improve the prediction performance. It is worth noting that our spatial positional encoding performs better in a scene with more people, which is in line with our expectations.

We report the MPJPE with different decoder query input length on CMU-Mocap in Tab. 4. We successfully prove that using only a single pose as the query for the decoder instead of a sequence of motion could create a *bottleneck* to force the Transformer to learn to predict the future instead of repeating the existing motion. We further explore the suitable number of the Transformer layers. We conduct experiments using 1, 3, 6 and 9 layers respectively on CMU-Mocap. The results in Tab. 5 show that a 3-layer-Transformer is suitable for both shorter and longer prediction. Fur-

|          | 1-layer | 3-layer | 6-layer | 9-layer |
|----------|---------|---------|---------|---------|
| 1 second | 1.02    | **0.96**| 1.07    | 2.69    |
| 2 seconds| 1.71    | **1.57**| 1.79    | 4.71    |
| 3 seconds| 2.47    | **2.18**| 2.58    | 6.83    |

Table 5: MPJPE with different number of Transformer layers on CMU-Mocap.

|             | 1 second | 2 seconds | 3 seconds |
|-------------|----------|-----------|-----------|
| fixed length| **0.96** | 1.91      | 2.91      |
| ours        | **0.96** | **1.57**  | **2.18**  |

Table 6: MPJPE on CMU-Mocap of our method and fixed length input.

|          | All DCT | No DCT | Ours    |
|----------|---------|--------|---------|
| 1 second | 0.99    | 1.44   | **0.96**|
| 2 seconds| 1.65    | 2.43   | **1.57**|
| 3 seconds| 2.37    | 3.47   | **2.18**|

Table 7: MPJPE on CMU-Mocap. Best results are shown in boldface.

thermore, to prove the effectiveness of our strategy described in 3.3, we conduct an experiment using a fixed length (one second motion) as input to the encoders and we report the quantitative comparisons using MPJPE on CMU-Mocap in Tab. 6. It can be seen our strategy is helpful in reducing the accumulation error, especially for long-term prediction.

We conduct ablation study on the role of DCT [4, 44]. In our method, we apply DCT to the input of the local-range Transformer encoder and apply IDCT to the output of the decoder. Because DCT can provide a more compact representation, which nicely captures the smoothness of human motion, particularly in terms of 3D coordinates [44]. We train a model which we apply all the input with DCT and apply IDCT to the output of the decoder, namely "All DCT". And we further train a model without any DCT or IDCT, namely "No DCT". We show the MPJPE on Mocap [1] in Tab. 7. We prove our design is more suitable for this task. Though DCT is quite helpful when predicting the human motion, from the experiment we find that for modeling the human interaction, position-based representation is more suitable.

## 4.5 Analysis of Attention

In this section, we discuss the attention results of each person in the decoder. In our model, the person in the last frame will query his/her past motion as well as the history motion of everyone. We give a

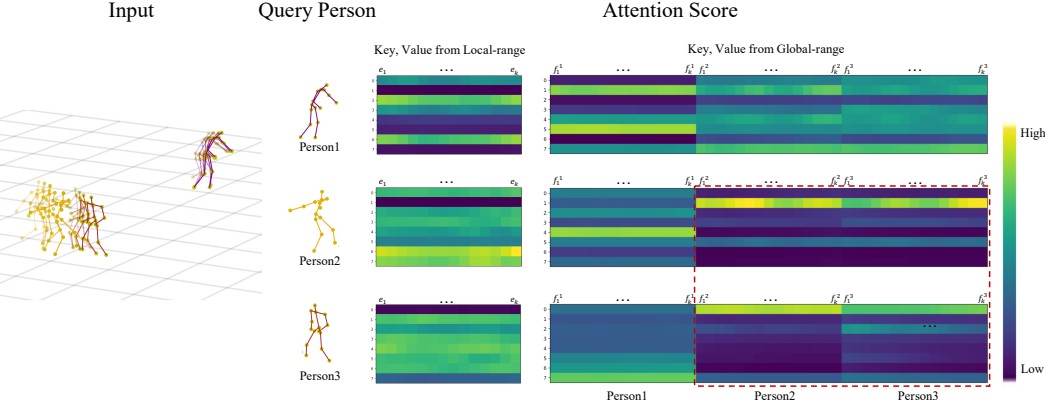

Figure 5: We show the input sequence, query person and the corresponding attention scores in the decoder. The dotted red box shows that the two interacting people have a more similar attention score distribution. Darker pose color represents the further in future.

visualization of the attention score in the first layer of the decoder in Fig. 5. On the left we show the history motion of three persons in the scene. On the right, we show different person's query and how the attention score looks like in the decoder. We separately normalize the scores for keys and values from the local-range encoder and the global-range encoder. Since we use a single frame $x_k$ to query, the query $q$ is only a vector, so is the attention score. We concatenate the attention score vectors from each head together and draw them as a matrix. Each row represents a head for a person and each column represents different persons at different time steps. Keys and values are from $[e_1, ..., e_k]$ and $[f_1^1, ..., f_k^1, ..., f_1^3, ..., f_k^3]$ respectively.

Fig. 5 shows that different query person could automatically weight different people at different time. Furthermore, we found that people in different interaction groups in a scene will have more similar attention score distributions when querying the outputs from the global-range encoder. This shows that our method has the ability to automatically divide the persons into different interaction groups in the decoder without any label of this. On the other hand, querying the outputs from the local-range encoder usually have a relatively fine-grained results. Specifically, the color changes more frequently. When querying outputs from the global range encoder, the color is relatively consistent in one head. This is consistent with our statements about the dense sampling and sparse sampling of the two encoders in Sec. 3.2.

## 5 Conclusion

In this paper, we propose a novel framework to predict multi-person trajectory motion. We design a Multi-Range Transformers architecture, which encodes both individual motion and social interaction and then outputs a natural long-term motion with a correspond pose as the query in the decoder. Compared with previous methods, our model can predict more accurate and natural multi-person 3D motion.

**Broader Impact.** The original intention of our research is to protect people's safety in surveillance systems, and collision avoidance for robotics and autonomous vehicles. However, we remain concerned about the invasion of people's privacy through the human motion and behavior. Since we do not use the image of a person as any input, it will not be easy to obtain the identity information of a specific person and we are pleased to see this. But we are still concerned about whether it is possible to identify a person based solely on the person's skeletons and motions.

## 6 Acknowledgments and Funding Transparency Statement

This work was supported, in part, by grants from DARPA LwLL, NSF CCF-2112665 (TILOS), NSF 1730158 CI-New: Cognitive Hardware and Software Ecosystem Community Infrastructure (CHASE-CI), NSF ACI-1541349 CC*DNI Pacific Research Platform, and gifts from Qualcomm, TuSimple and Picsart.

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
