# Supplementary Material: Multi-Person 3D Motion Prediction with Multi-Range Transformers

**Jiashun Wang**[1]    **Huazhe Xu**[2]    **Medhini Narasimhan**[2]    **Xiaolong Wang**[1]
[1]UC San Diego        [2]UC Berkeley

We provide more details about datasets, ablation study, user study and the analysis of the attention in the supplementary. We also provide a supplementary video to better visualize the results.

## 1 Dataset

Our goal is to predict accurate and high-quality multi-person 3D human motion, so for the data selection, we hope to train on a dataset with smoother movements (CMU-Mocap [1]), and then test the generalization capability of our model on datasets obtained through the estimation [7, 6, 4]. There are more persons in Panoptic [4] in each scene. We take this into consideration and use it to augment the training data for the setting with more persons. We sample 6k sequences containing multi-person human motion for the setting with small number of people and setting with large number of people. Each sequence lasts for 4 seconds. For evaluation, we sample 800 sequences from CMU-Mocap and Mix1. Due to the limiting amount of data, we sample 100 sequences from MuPoTS-3D, 200 sequences from 3DPW and 100 sequences from Mix2. Each sequence lasts for 4 seconds. Considering that these data have different skeleton representations, e.g. the number of the joints, we select 15 joints they have in common as training and test data. In Fig 1, we visualize the results before and after pre-processing. At the same time, in order to ensure that these people can appear in the same scene, we scale the human skeletons from different datasets such that each person is between $1.5m - 2m$ high. In a scene with a small number of people, we randomly place each group of people in a $25m^2$ square on the x-y plane. For scenes with large number of people, we randomly place each group of people in a $100m^2$ square on the x-y plane.

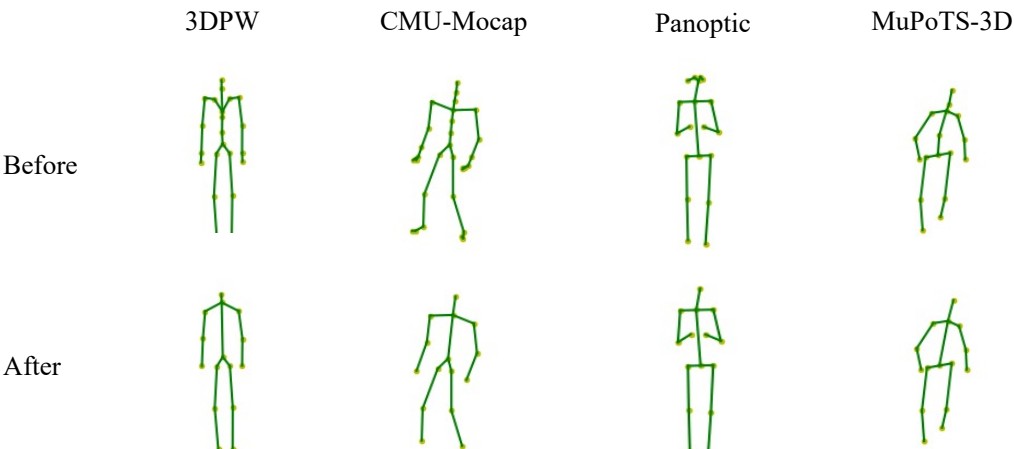

Figure 1: Visualization of skeletons before and after the pre-processing.

35th Conference on Neural Information Processing Systems (NeurIPS 2021).

## 2 UI for user study

Following [9, 8], we perform a user study using Amazon Mechanical Turk. We generate 50 4-second motions for each dataset respectively and ask the users to give a score between 1 (strongly not natural) and 5 (strongly natural) to each result. We give a screen shot of the interface of our user study in Fig. 2.

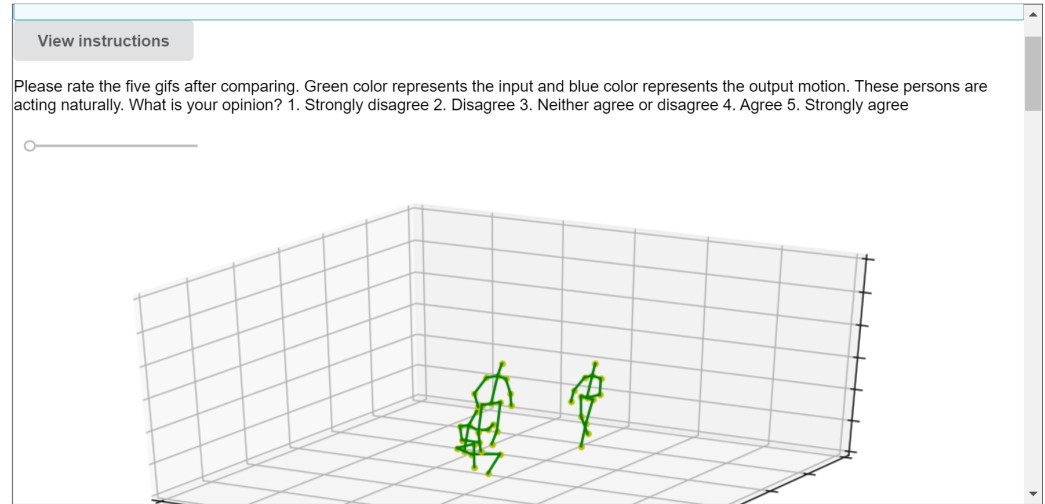

Figure 2: The interface of of the user study on AMT.

## 3 Implementation details of method for comparison

Since there is no public implementation of SocialPool [2], we introduce the details of our implementation in the experiment. We use the same training data as our method. And same as our method, we use the offset representation for input and output. We apply a GRU [3] encoder to encode each person's history motion sequence. We then apply maxp ooling to all persons' feature and get a social feature. We concatenate the corresponding feature and the social feature and send it to GRU decoder to output each person's future motion sequences. We use a batch size of 32 and a learning rate of $1 \times 10^{-4}$. We use Adam [5] as the optimizer.

## 4 Attention visualization

We give a visualization of the attention score of the global motion in the decoder in Fig. 3. We use different color to represent the people in different interaction groups. Person 1 and person 4 are alone and not interacting with anyone. It can be seen that the attention scores of the people belonging to the same interaction group have a more similar distribution. We apply our Spatial Positional Encoding (SPE) to the global-motion feature and thus different from previous transformer-based methods, the value inputs for the multi-head attention have already got connections with the query input for the multi-head attention. Thus the attention score does not fully reflect the relevance between the query and value, since a more relevant value could be larger after applying our SPE. However, we argue that the similarity of the distribution can reflect whether they are interacting. More importantly, our method computes attention scores not only based on the distance among the people, but also takes into consideration of people's behaviors. For example, in Fig. 3, although person 1 is next to person 10 and person 11, it has quite different attention score distribution from person 10 and person 11.

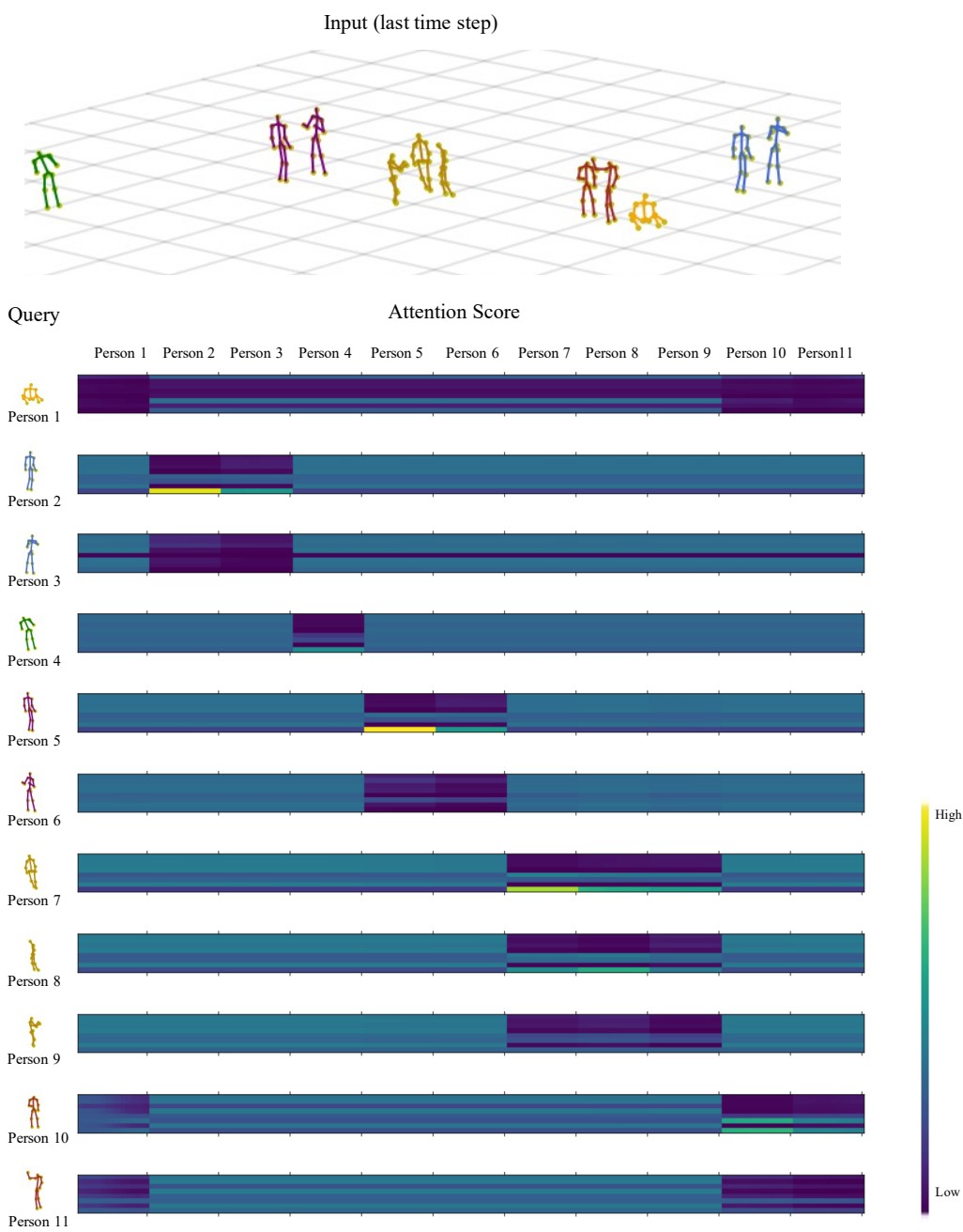

Figure 3: We show the last time step of the input on the top and below we show the query and the attention score. People in the same interaction group have a more similar distribution of the attention score.