# OpenReview forum: "Multi-Person 3D Motion Prediction with Multi-Range Transformers"
_NeurIPS.cc/2021/Conference — NeurIPS 2021 Poster_

### Official Review · Reviewer_uWwa · 2021-07-03

**Rating:** 6
**Confidence:** 4

**Summary:**

This paper proposes a transformer-based model to predict future 3D poses of a group of people, given 3D poses from previous observations. The authors explicitly model local interactions (ie dependencies between poses at previous time steps for a given person) and global interactions (dependencies between different people in the scene), and obtain state-of-the-art results on 3 different datasets. The transformer model is an encoder-decoder type structure, where the present and past poses are encoded, and the query to the decoder is the pose at the current frame. The model then predicts 3D poses for T future frames.

**Limitations And Societal Impact:**

Both of these are barely described in the main paper. The checklist says they are detailed in the supplementary, but I did not see it there.

**Main Review:**

The method proposed by the authors, using an encoder-decoder structure of transformers (similar to that of the original transformer for machine translation), is a good way of modelling temporal dependencies in 3D pose of a single person, and also interactions with other people in the scene. The transformer-based architecture can also naturally handle a variable sequence length at the input.

The authors use two encoders, one to model local interactions, and another to model global interactions between all people in the scene. The "Global range transformer encoder" has N times more input tokens than the local transformer, where N is the number of people. Therefore, since the "Global transformer" is a superset of the "Local transformer", is the local transformer strictly necessary? And would any potential benefits of having a "Local + Global" transformer combination disappear as the training dataset size increases? Results from other large-scale transformer papers [17] suggest that additional inductive biases in the network design becomes less important with larger dataset and model sizes. I would appreciate the author's comments on this.

An interesting choice made by the authors was to progressively increase the input sequence length to the model during inference. Ie, the input pose, and all previous predictions made in the sequence are used as the input to the encoder of the model. It would be good to ablate the effect of this design choice: What if the input sequence length is always fixed (except the beginning of the sequence)?

The authors also include some qualitative results showing that the decoder groups together groups of interacting people. However, the authors use a synthetically created dataset where they mix pose trajectories from different sources (ie CMU-Mocap and Panoptic). Therefore, it is easy to distinguish these groups based on the positional embedding alone, as the "mixed samples" are far apart in 3D space. These results would be more interesting to me, if the model was able to distinguish groups of people from the same original scene. Right now, it appears that the model is distinguishing people from different dataset splits instead.

The model produces good results in terms of MJPJE accuracy. However, future prediction by definition can have multiple possible outcomes, and the proposed method always predicts a single prediction, rather than diverse ones (ie [36]).

It would also be good to discuss this concurrent work in the final version (although I understand that it is too recent for the authors to compare to)

M Petrovich et al. Action-Conditioned 3D Human Motion Synthesis with Transformer VAE. arxiv 2104.05670. 2021

There are also a couple of typos in the text

Line 23: "ma" -> "may"
Line 24: "response accordingly" --> "respond accordingly"
Line 70 and 103: "in the world coordinate" -> "in world coordinates" / "in the world coordinate frame"
Line 137: "plus the sinusoidal embedding" --> "add the sinusoidal embedding"



Overall, I am quite borderline with this paper. It shows that a transformer-based encoder-decoder model is effective for the task of 3D future prediction. However, it does not appear that original to me, given the success of transformers in so many different domains. And the proposed model is fairly similar to the encoder-decoder model for machine translation in the original transformer paper. Although the authors perform explicit modelling of "local" and "global" interactions, I do not think they have really adapted the transformer architecture specifically to the future prediction task by for example, generating diverse outputs, or by conditioning the future prediction on some input.

**Time Spent Reviewing:**

4

---

> ### Author Response · Authors · 2021-08-10
> **Response to reviewer uWwa**
>
> Dear reviewer uWwa, thank you for your detailed and thorough review. We have made a general response to provide further discussion and insights into our method. We sincerely thank you for the advice and pointing out the typos. We will revise them in the future version.  In the following, we seek to address each of your concerns.
>
> **Q**: *“Therefore, since the "Global transformer" is a superset of the "Local transformer", is the local transformer strictly necessary? And would any potential benefits of having a "Local + Global" transformer combination disappear as the training dataset size increases? Results from other large-scale transformer papers [17] suggest that additional inductive biases in the network design becomes less important with larger dataset and model sizes. I would appreciate the author's comments on this.”*
>
> **A**: For detailed discussion about local-range and global-range encoders, please refer to our answer on “**Why local-range and global-range**” in the general response. Our two encoders are designed for modeling two kinds of information and we import our understanding of the problem as the inductive bias to the model for better optimization. While [17] show large-scale transformer works for large-scale image recognition, our problem focuses on 3D data which has a limited scale, making model design much more important.
>
> ---
>
> **Q**: *“The authors also include some qualitative results showing that the decoder groups together groups of interacting people. [...] it is easy to distinguish these groups based on the positional embedding alone, as the "mixed samples" are far apart in 3D space. These results would be more interesting to me, if the model was able to distinguish groups of people from the same original scene.”*
>
> **A**: In Fig.3 in the supplementary material, we give an example showing that people are not only distinguished by distance (spatial positional encoding). For example, person 1 is quite close to person 10 and person 11, but the distributions of person 10 and person 11 are similar while person 1 is not. Our method could not only take the distance information but will also analyze the specific motion information. Please see this [link](https://raw.githubusercontent.com/AnonymousAut/Anonymous-Result/gh-pages/rebuttal/att.jpg) for a shortcut to this example.
>
> ---
>
> **Q**: *“The model produces good results in terms of MJPJE accuracy. However, future prediction by definition can have multiple possible outcomes, and the proposed method always predicts a single prediction, rather than diverse ones (ie [36]).”*
>
> **A**: Please refer to our answer on “**Multiple possibility prediction**” in the general response.
>
> ---
>
> **Q**: *“It would also be good to discuss this concurrent work in the final version.”*
>
> **A**: Thank you for the advice and we will discuss this and more concurrent works in the final version.
>
> ---
>
> **Q**: *“An interesting choice made by the authors was to progressively increase the input sequence length to the model during inference[...]. It would be good to ablate the effect of this design choice: What if the input sequence length is always fixed (except the beginning of the sequence)?”*
>
> **A**: Empirically, we think progressively increasing the input sequence length during training and inference is a good choice, since the transformer can deal with variable length input. We perform an experiment always using one second motion as input. We report the quantitative results on CMU-Mocap below. We prove progressively increasing the input sequence could largely improve long-term prediction. We think letting the model see motions in the beginning is a very helpful strategy to reduce the error accumulation.
>
> | | 1 second | 2 seconds | 3 seconds
> --- | --- | --- | ---
> fixed length | 0.96 | 1.91 | 2.91
> ours | 0.96 | 1.57 | 2.18
>
> ---
>
> **Q**: *“However, it does not appear that original to me, given the success of transformers in so many different domains. And the proposed model is fairly similar to the encoder-decoder model for machine translation in the original transformer paper. Although the authors perform explicit modelling of "local" and "global" interactions, I do not think they have really adapted the transformer architecture specifically to the future prediction task[...].”*
>
> **A**: Transformer has shown its success in many different domains. We also found it useful in modeling the motion sequence. However, a simple encoder-decoder transformer model could not directly solve the multi-person 3D motion prediction task very well. In particular, we need to model one’s motion and also the interaction with others. We need to predict a natural motion, which also involves interacting with other people in a natural way. We design our multi-range transformers to solve it, and we discuss more about why we use local-range and global-range encoders in “**Why local-range and global-range**” in the general response.
>
> At the same time, predicting multi-person 3D motion is a relatively under-explored problem. We achieve large improvements over existing methods in both quantitative and qualitative, which is helpful to the community.
>
> ---
>
> **Q**: Limitations and Social Impact.
>
> **A**: Since we do not use the image of a person as any input, it will not be easy to obtain the identity information of a specific person and we are pleased to see this. But we are still concerned about whether it is possible to identify a person based solely on the person’s skeletons and motions.

---

> > ### Comment · Reviewer_uWwa · 2021-08-28
> > **Update after rebuttal**
> >
> > Thank you for the rebuttal.
> >
> > I think it addresses my concerns, and those of the other reviewers, quite well. As a result, I am upgrading my rating.
> >
> > I do, however, think that Reviewer 7u8M made a valid comment about comparisons to [2]: The authors should be more explicit in the experiments section that they have reimplemented [2]. I also encourage the authors to release their code and models (both for their proposed method and the baselines) after acceptance to make it easier for follow-up work to compare to them.

---

### Official Review · Reviewer_c8qX · 2021-07-08

**Rating:** 7
**Confidence:** 3

**Summary:**

--- Summary

This paper proposes a model for multi-person 3D motion prediction. Different from the previous works, the proposed model uses the Transformer architecture and is able to better model the multi-person social interactions. The model consists of a local-range encoder to encode the motion of each individual person, a global-range encoder to encode the social interactions, and a transformer decoder to generate the trajectory predictions. And it also has a discriminator module to make the generated trajectories more natural. The proposed model is trained with both the reconstruction loss and the discriminator loss. The evaluations were performed on multiple datasets including CMU-Mocap, Panoptic, MuPoTS-3D, and 3DPW, and the proposed model outperforms the baselines by 20% ~ 30%.

**Ethical Concerns:**

No.

**Limitations And Societal Impact:**

Yes, the authors have adequately addressed the limitations and potential negative societal impact of their work.

**Main Review:**

--- Strengths

- The evaluation is thorough. The proposed model was compared against multiple state-of-the-art baselines on multiple datasets, and it was shown to outperform the baselines by a large margin. The authors also performed a few ablation studies to show that the proposed design components are necessary. There was also a user study performed with Amazon Mechanical Turk that compared the quality of the predictions.

- The paper is well-written.


--- Issues and suggestions

- It's interesting to have the user study that quantitatively compares the quality of the predictions, but all models compared (and including the ground-truth) seem to end up having very close scores, so maybe the design of this study can be improved?


--- Other comments

- Typo in Line 22: "ma".


**Time Spent Reviewing:**

2

---

> ### Author Response · Authors · 2021-08-10
> **Response to reviewer c8qX**
>
> Dear reviewer c8qX, thank you for your detailed and thorough review. We have made a general response to provide further discussion and insights into our method. We sincerely thank you for the advice and pointing out the typos. We will revise them in the future version. In the following, we seek to address each of your concerns.
>
> **Q**: *“It's interesting to have the user study that quantitatively compares the quality of the predictions, but all models compared (and including the ground-truth) seem to end up having very close scores, so maybe the design of this study can be improved?”*
>
> **A**: Since we let users score 1, 2, 3, 4 or 5, we find the users intend to give scores mostly around 3 or 4. This may lead to relatively small differences between the methods but in this case the average score is still statistically meaningful to compare. Ground truth results for some datasets are obtained by pose estimation techniques so they are not perfect as well. More visual results and comparisons can be found in our supplementary video and website: https://anonymousaut.github.io/Anonymous-Result/. These can also show our improvement over other methods. We will redesign our interface for ranking the approaches in our revision of the paper.

---

> > ### Comment · Reviewer_c8qX · 2021-08-21
> > **Thank you for your responses**
> >
> > Thank you for your responses.

---

### Official Review · Reviewer_BvU8 · 2021-07-13

**Rating:** 7
**Confidence:** 4

**Summary:**

- The paper proposes to tackle the problem of multi person 3D motion prediction.
- Given the trajectories of multiple people in the scene, the model reasons about both long and short term context using separate transformer based encoders for both.
- A transformer based decoder then takes a single pose at a time step for each person and uses encoded features to predict the future trajectory.
- Experiments are performed on the CMU-Mocap, MuPoTs-3D and 3DPW dataset. and the results show improvements over state-of-the-art.
- Qualitative visualizations show that the model can uncover behaviors, interactions and group people in crowds.


**Limitations And Societal Impact:**

- The authors have adequately discussed the limitations and societal impact of their work.

**Main Review:**

Strengths:
- The problem is properly motivated.
- Transformers for the problem, are natural given that it can handle variable length input and handle long range temporal dependencies.
- The idea of combining global and local in the proposed way is interesting. Not all global context might be important for all pose sequences, the proposed approach helps tackle that.
- The proposed approach shows considerable improvement over state-of-the-art approaches
- The prediction visualizations when groups of people were involved were quite interesting and the accompanying video in the supplementary was helpful.
- The experimentation is thorough and includes a user study.

Concerns:
- DCT : It is not clear to me why DCT was used for this task. The paper [44] uses DCT to encode temporal information and then works with the DCT bases for input to a graph model. In this paper, the transformer encoder already captures the temporal information. Was it merely an empirical observation (as given in the supplementary) ?
- L136-137 : Was the position embedding added to the concatenated embedding or to the output of the linear layer ?
- Global-range transformer : Should you have an input dimension which differentiates embeddings of the different people from each other ? Spatial position encoding could potentially be useful for this but I am curious why that encoding was added after the encoder ?
- Will the calculation of SPE scale ? Since it requires computing distance between every pair.
- While the user study does indicate a slight improvement compared to the others, the difference seems to be small. The GT values seem small too. Is that expected ?
- Did the authors experiment with any data augmentations ?

Other suggestions :
- Typo : L23 : ‘ma’
- Rephrase : L 120-121
- Typo : L134 : ‘contact’ -> ‘concatenate’ ?
- Rephrase : L167-168




**Time Spent Reviewing:**

4

---

> ### Author Response · Authors · 2021-08-10
> **Response to reviewer BvU8**
>
> Dear reviewer BvU8, thank you for your detailed and thorough review. We have made a general response to provide further discussion and insights into our method. We sincerely thank you for the advice and pointing out the typos. We will revise them in the future version. In the following, we seek to address each of your concerns.
>
> **Q**: *“DCT : It is not clear to me why DCT was used for this task.”*
>
> **A**: Please refer to our answer on “**Usage of DCT**” in the general response.
>
> ---
>
> **Q**: *“L136-137 : Was the position embedding added to the concatenated embedding or to the output of the linear layer?”*
>
> **A**: It is added to the output of the linear layer.
>
> ---
>
> **Q**: “Global-range transformer: Should you have an input dimension which differentiates embeddings of the different people from each other? Spatial position encoding could potentially be useful for this but I am curious why that encoding was added after the encoder?”
>
> **A**: We do not provide the identity information as inputs since we find the pose and trajectory information are already enough for distinguishing different people.
> The spatial position encoding is added after the encoder because: When we use different persons’ pose as the query for the decoder (to predict different persons’ motion), we need to use the spatial positional encoding to specify the person. If it is added before the encoder, we need to perform forward propagation with the global encoder for each query person individually. If it is added after the encoder, we only need to compute the global feature once, which saves a lot of computation.
>
> ---
>
> **Q**: *“Will the calculation of SPE scale ? Since it requires computing distance between every pair.”*
>
> **A**: Yes it is scaled now. We perform an exponent to the negative of the distance. So all the numbers are from 0 to 1.
>
> ---
>
> **Q**: *“While the user study does indicate a slight improvement compared to the others, the difference seems to be small. The GT values seem small too. Is that expected?”*
>
> **A**: Ground truth results for some datasets are obtained by pose estimation techniques so they are not perfect as well. Since we let users score 1, 2, 3, 4 or 5, we find the users intend to give scores mostly around 3 or 4. This may lead to relatively small differences between the methods but in this case the average score is still statistically meaningful to compare.  Ground truth results for some datasets are obtained by pose estimation techniques so they are not perfect as well. More visual results and comparisons can be found in our supplementary video and website: https://anonymousaut.github.io/Anonymous-Result/. These can also show our improvement over other methods. We will redesign our interface for ranking the approaches in our revision of the paper.
>
> ---
>
> **Q**: *“Did the authors experiment with any data augmentations?”*
>
> **A**: We perform data augmentation when we mix the datasets, since we randomly sample motions from different datasets and randomly put them into one plane ground. And all the methods are trained using the same data.

---

> > ### Comment · Reviewer_BvU8 · 2021-08-31
> > **Thanks for the responses**
> >
> > Thanks for your responses! I think the rebuttal adequately addresses my concerns. Regarding some other shared concerns:
> >
> > - Use of local & global range : I found the provided arguments to be convincing. The qualitative results seem to corroborate them.
> > - Multiple possibilities : I think this paper has enough content and multiple possibility could be seen as an extension.
> > - Grouping of people based on position alone: This was an interesting analysis but become less significant if its easy to distinguish groups based on position embedding alone as rightly pointed out by reviewer uWwa. The author provide an example of where this is not a case. Some more examples/analysis here would have helped. But, I don't think this is a major weakness of this work.
> > - As pointed out by Reviewers 7u8M and uWwa, the authors should make the experimental setting more clear with regards to [2]
> >
> > I think that the paper is well motivated, with an interesting approach with thorough qualitative and quantitative analysis.
> > Based on the other reviews, author response and comments from other reviewers, I will retain my rating.

---

### Official Review · Reviewer_i6Ex · 2021-07-15

**Rating:** 7
**Confidence:** 4

**Summary:**

This paper proposes a Transformer-based deep learning architecture to predict multi-person 3D motion trajectories. To better model the social influence of neighbors, both a local-range (individual motion) and a global-range (social interactions) Transformer encoders are used.
The proposed framework is benchmarked on several 3D motion prediction datasets, namely CMU-Mocap, Panoptic, MuPoTS-3D, and 3DPW (and some combinations of these). The results show an improvement w.r.t. the select baselines of 20-30%. From a qualitative point of view, the model predictions seem smoother and more natural compared to previous efforts. Furthermore, the model is able to predict many people at the same time, and different social groups appear to cluster together when examining the social attention scores.

**Limitations And Societal Impact:**

I believe limitations and potential negative societal impact of this work have been adequately addressed by the authors.

**Main Review:**

The submission presents a state-of-the-art model for 3-D human motion trajectory prediction. More specifically, the model uses two encoders and one decoder based on Transformers and a discriminator consisting of fully-connected layers. I think the overall quality of the paper is good and both the architecture and results are systematically presented.

The comparison with previously published results is properly done, and the proposed method is able to outperform state-of-the-art performance by a noticeable margin. The two considered experimental settings, namely generalization and fusion, allow the reader to understand the impact of considering more than 2/3 people as well as the generalization capability of the model.

However, I have some concerns regarding several statements reported in the paper. For example:
- the authors states in the abstract that the model 'also generates diverse social interactions'. I think this aspect should be better investigated (or explained) since it is not clear from the qualitative and quantitative results. What does 'diverse' mean in this case? 'Different' among the people in the scene?
- Then, regarding the second experimental setting, the authors fuse several scenes from different datasets but they use absolute coordinates. How the different coordinates have been taken into account, if they are different? And, most importantly, people are involved in different actions and fusing them will
result in not natural interactions. Is my interpretation correct?
- In Section 3, it is stated that 'x^k_n contains both the trajectory and the pose information'. It is not clear to me the difference since the dimension of this vector is '3J' and not '3J+1' as it should be with the addition of the trajectory information.
- Absolute coordinates are then transformed into relative motion (x_(i+1) - x_i) in this work while, as stated, previous works use joint positions centered at the origin.  Is this not the same thing? If not, why?

Besides the above concerns, I report some minor comments below:
- Page 1, line 23: 'ma' should be 'may';
- Page 1, line 25: 'response' should be 'respond';
- Page 1, line 27: 'Such a model will need have the following'. This sentence should be correctly rephrased;
- Page 4, line 130: 'update' is repeated two times;
- Page 5, line 160: 'this single pose to as'. 'to' should be removed;
- Page 6, line 210: ', We sample' should be ', we sample';
- Page 7, line 272: 'cross' should be 'across';
- It would be better, in my opinion, to report the number of observed and predicted time-steps in section 3 (Method) since it is a very important detail of the task;
- The authors perform experiments considering 9-15 people. It would be nice to see a qualitative example of this setting in the paper.

**Time Spent Reviewing:**

7

---

> ### Author Response · Authors · 2021-08-10
> **Response to reviewer i6Ex**
>
> Dear reviewer i6Ex, thank you for your detailed and thorough review. We have made a general response to provide further discussion and insights into our method. We sincerely thank you for the advice and pointing out the typos. We will revise them in the future version. In the following, we seek to address each of your concerns.
>
> **Q**: *“the model 'also generates diverse social interactions'. [...] is not clear from the qualitative and quantitative results. What does 'diverse' mean in this case?”*
>
> **A**: We clarify that 'diverse' means that our method can model different kinds of interactions instead of a fixed number of motion categories. Conditioning on different interaction inputs, our method can correspondingly predict different interactions and motions, while previous approaches might still predict similar motions with different inputs. We will make this more clear in our future revision.
>
> ---
>
> **Q**: *“the authors fuse several scenes from different datasets but they use absolute coordinates. How the different coordinates have been taken into account, if they are different? [...] people are involved in different actions and fusing them will result in not natural interactions.”*
>
> **A**: We describe how to perform fusion in section 1 in our supplementary material. We will scale and normalize the human skeletons into the same coordinate space before fusion. We place the skeletons randomly into a plane ground. We will not place different interaction groups in the exact same place but there is a possibility that people who are not interacting are close to each other. We have provided one example in Fig.3 in supplementary material.
>
> ---
>
> **Q**: *“In Section 3, it is stated that 'x^k_n contains both the trajectory and the pose information'. It is not clear to me the difference since the dimension of this vector is '3J' and not '3J+1' [...]”*
>
> **A**: We are using the absolute coordinates. The xyz coordinates of each joint contain both the trajectory information and pose information. Thus the vector size of ‘3J’ contains the information of both trajectory and pose.
>
> ---
>
> **Q**: *“Absolute coordinates are then transformed into relative motion (x_(i+1) - x_i) in this work while, as stated, previous works use joint positions centered at the origin. Is this not the same thing? If not, why?”*
>
> **A**: It is not the same. Previous works did not consider the trajectory movement (global movement) and only considered the pose change (relative positions between pose joints).  For example, if a person is jumping forward, previous works may only model the ‘jump’ pose without modeling ‘forward’. And in our work, (x_(i+1) - x_i) contains the information of both ‘jump’ and ‘forward’.
>
> ---
>
> **Q**: *“It would be better, in my opinion, to report the number of observed and predicted time-steps in section 3.”*
>
> **A**: Thank you for your advice. Our method takes 1 second history motion (k=15 time steps) as input and recursively predicts the future 3 seconds (k=45 time steps). We will adjust section 4.2 and add this in section 3.
>
> ---
>
> **Q**: *“The authors perform experiments considering 9-15 people. It would be nice to see a qualitative example of this setting in the paper.”*
>
> **A**: We have provided examples showing this case in our supplementary video and website: https://anonymousaut.github.io/Anonymous-Result/

---

### Official Review · Reviewer_7u8M · 2021-07-16

**Rating:** 3
**Confidence:** 4

**Summary:**

The paper proposes a learning framework based on transformers for 3D human body motion prediction considering the motion and interaction of other people in the scene. The proposed approach contains a local-range transformer encoder to encode each individual motion and a global-range transformer encoder for encoding social interactions. Then a transformer decoder is used to predict the future motion for each person. The results are evaluated on three datasets, CMU-Mocap, MuPoTS-3D, and 3DPW  and compared against few state-of-the-art methods.

**Limitations And Societal Impact:**

The authors attempted to address the limitation and potential negative social impact of their work on page 9.


**Main Review:**

1- Clarity: The main problem, motivation and the idea of the paper including its technical details are very clear. Though, the paper contains quite a few typos and grammatical errors/

2- Contribution: the paper does not offer any novel theoretical and practical contribution. The method uses some arbitrary architecture variants of transformer encoders followed by a transformer decoder to predict the 3D human body skeleton of each person. The idea of social interaction for predicting human motion is not new, even in the context of the human skeleton body pose (e.g. 2 and 3). Transformers has been already used to incorporate human interaction for the human trajectory forecasting problem, eg Yu et al, "Spatio-Temporal Graph Transformer Networks for Pedestrian Trajectory Prediction", ECCV 2020.

3- Experiments: The results of the proposed method have only been compared against a few out of many existing frameworks on human body skeleton prediction and only tested on few arbitrary datasets. Also, no details for the implementation (or hyper-parameters) of the competing methods are provided in the main text or supp material file. It is also not clear how these methods are compared because the frameworks [43] and [44] are designed to predict human body skeleton motion only (no trajectory motion is considered or predicted), while the evaluated datasets and framework [2] consider both tasks jointly. I couldn't find any public implementation for the framework [2]. it is very important to elaborate the implementation and hyperparameters details of competing methods to ensure a fair comparison. I could also see there exist a benchmark named SoMoF which aims to provide a benchmark for the same problem this paper targeted.  This approach could be evaluated on this benchmark instead. Human body pose and motion are not only affected by the other human body motion but also the objects and obstacles in the scene. several human body motion forecasting frameworks have been already proposed to model these physical interactions e.g. Cao et al. Long-term human motion prediction with scene context. ECCV, 2020 and the comparison with these relevant works fare missing,

**Time Spent Reviewing:**

2

---

> ### Author Response · Authors · 2021-08-10
> **Response to 7u8M**
>
> Dear reviewer 7u8M, thank you for your detailed and thorough review. We have made a general response to provide further discussion and insights into our method. In the following, we seek to address each of your concerns.
>
> **Q**: *“The method uses some arbitrary architecture variants of transformer encoders followed by a transformer decoder [...].”*
>
> **A**: We respectfully disagree with the reviewer using the word “arbitrary” to describe our method. We emphasize the problem of multi-person 3D motion and trajectory prediction is a relatively under-explored area. We conducted extensive studies on both the local-range encoder and the global-range encoder and showed their necessity and effectiveness for this problem. Thus our method design is supported by experimental evidence rather than “arbitrary”.
>
> ---
>
> **Q**: *“The idea of social interaction for predicting human motion is not new [...]. Transformers has been already used to incorporate human interaction for the human trajectory forecasting problem, eg Yu et al. Spatio-Temporal Graph Transformer Networks for Pedestrian Trajectory Prediction, ECCV 2020.”*
>
> **A**: We did not claim that we proposed the problem of social interaction for predicting human motion. We discussed the previous literature in detail in our related work. While Transformer has been applied on trajectory forecasting like Yu et al, it did not model the human pose which has much less complexity in prediction. Our special design of transformers on the other hand, allows for long-term human trajectory and pose prediction, which has not been studied before. This is also acknowledged by **Reviewer MkB1** on Originality: *“As far as I know this method is new for the tasks of multiple person motion prediction.”* and **Reviewer BvU8** on Strengths: *“The idea of combining global and local in the proposed way is interesting. Not all global context might be important for all pose sequences, the proposed approach helps tackle that.”*
>
> ---
>
> **Q**: *“The results of the proposed method have only been compared against a few out of many existing frameworks on human body skeleton prediction and only tested on few arbitrary datasets.”*
>
> **A**: We respectfully disagree with the reviewer using the word “arbitrary” again for describing our dataset selection. We emphasize again that this problem is relatively under-explored and we have tried all the possible ways to find and construct available datasets (**across 4 datasets** in total) and as many as baselines (**3 baselines and 3 more variants of our method**). Our experiment evaluation is acknowledged by **Reviewer c8qX**: *“The evaluation is thorough. The proposed model was compared against multiple state-of-the-art baselines on multiple datasets, and it was shown to outperform the baselines by a large margin.”* and by **Reviewer i6Ex**: *“The comparison with previously published results is properly done, and the proposed method is able to outperform state-of-the-art performance by a noticeable margin. ”*
>
> ---
>
> **Q**: *“no details for the implementation (or hyper-parameters) of the competing methods are provided in the main text or supplementary  material file. [...] [43] and [44] are designed to predict human body skeleton motion only [...] I couldn't find any public implementation for the framework [2].”*
>
> **A**: We have kept the original settings of [43] and [44] and apply them on our datasets for training. We have explained in LN 241 in the paper that the architectures in [43, 44] actually allow modeling the pose and trajectory jointly. While the code of [2] is not available, we managed to reproduce their approach in their own datasets and applied the same code to our dataset. We believe these are the maximum efforts one could have tried to make comparisons. We argue a paper should not be rejected because the code from a previous paper is not released. We will include more details of implementation in our future revision.
>
> ---
>
> **Q**: *“I could also see there exist a benchmark named SoMoF which aims to provide a benchmark for the same problem this paper targeted. This approach could be evaluated on this benchmark instead.”*
>
> **A**: SoMoF is proposed with [2], which we have compared. We want to solve a relatively long-term (3 seconds) motion prediction problem. Currently we do not take the image context as consideration, we are focusing on human motion and multi-person interaction, rather than human-scene interaction. At the same time, we try to model more persons in one scene,
>
> ---
>
> **Q**: *“several human body motion forecasting frameworks have been already proposed to model these physical interactions e.g. Cao et al. Long-term human motion prediction with scene context. ECCV, 2020 and the comparison with these relevant works fare missing.”*
>
> **A**: We find this comment unreasonable and the request is unfair. We are tackling the problem of multiple-person interaction and prediction, and the reviewer requested us to solve another task and compare a paper on single person prediction with person-scene interaction. We argue that our problem is an important problem, which is agreed by other reviewers. We do not think the request to force us to solve another problem is a proper and fair comment.

---

> > ### Author Response · Authors · 2021-09-02
> > **Please let us know whether you have additional questions**
> >
> > Dear Reviewer,
> >
> > We have provided more results and explanations based on your review. Since today is the last day for discussion, can you please go over them and let us know whether you have additional questions or not?
> >
> > Thank you very much.

---

### Official Review · Reviewer_MkB1 · 2021-07-20

**Rating:** 6
**Confidence:** 3

**Summary:**

This paper describes a model for multi-person motion prediction. It is based on the encode-decoder architecture. Both are constructed with self-attention transformers. The model takes in the absolution 3D coordinates of body joints from multiple persons and predict the future movement of the joints in 3D space for a timespan of 1-3seconds.

Additionally, the authors introduced several pactices in the model design including: 1) the use of the last seen frame in the observed as queries; 2) the use of DCT to encode joints; 3) The separation of a global encoder and a local encoder. The global encodes input of multiple persons while a local encoder encoder data from a single person. The model is then trained with both the reconstruction loss and a adversarial loss implemented via a Transformer based discriminator model.

The authors conducted experiments on human motion sequence datasets. Improved results over compared baselines are observed. The authors further provided analysis on the query design and number of transformer layers used. Qualitative analysis of the prediction is also provided.

**Limitations And Societal Impact:**

The authors has addressed the societal impact of the proposed method.

I didn't see much discussion about the limitation of the work.

**Main Review:**

## Originality

As far as I know this method is new for the tasks of multiple person motion prediction. There is a related field of object trajectory prediction but in that task mostly the trajectories of centers of mass are concerned while in this work each joint's location needs to be predicted.

## Quality
The model itself is described clearly. It achieves good improvement over the baselines. The authors have provided a few ablation studies of design choices.

There are several things that are not assessed as listed below:
- whether the local range Transformer is necessary given that the global transformer already sees all input data.
- the use of DCT as input encoding
- the combined prediction of absolute joint locations vs. center of mass trajectory + relative joint locations
Some of these items are claimed as motivation of design choices so they warrant a study.

The description of the discriminator is a little bit vague. One questions not answered is why it is not designed as a similar architecture as the prediction part. In experiments, it is only giving marginal improvements on most of the datasets (Table 1/2/3). So this seems either a missing opportunity or deadweight in the design. The authors please clarify.

 ## Significance
Being able to predict the motion of multiple person simultaneously is a nice and practical property of the proposed method. Based on this I feel it will be useful for practitioners. The idea of local vs. global in prediction is interesting but in the current form it lack thorough investigation.

## Additional questions

I have the following additional questions:

1. The MPJPE metric has no unit. Is it in mm, cm, m, or any other metric system?
2. Predicting future motion is by its natural non-deterministic. Here we are comparing the prediction with the "groundtruth", which is one possible instantiation among many possible futures. Is there any consideration in this aspect?


**Time Spent Reviewing:**

5

---

> ### Author Response · Authors · 2021-08-10
> **Response to reviewer MkB1**
>
> Dear reviewer MkB1, thank you for your detailed and thorough review. We have made a general response to provide further discussion and insights into our method. In the following, we seek to address each of your concerns.
>
> **Q**: *“Whether the local range Transformer is necessary given that the global transformer already sees all input data” and “The idea of local vs. global in prediction is interesting but in the current form it lack thorough investigation.”*
>
> **A**: We have compared the performance between our method and one without Local-range Transformer in the Tab.2 in the supplementary material. We also provide a [visual comparison](https://raw.githubusercontent.com/AnonymousAut/Anonymous-Result/gh-pages/rebuttal/local.gif)
> Local-range Transformer is not only helpful to predict a more accurate motion but also helps generate smoother motion. And we discuss more about why we use local-range and global-range encoders in “**Why local-range and global-range**” in the general response.
>
> ---
>
> **Q**: *“the use of DCT as input encoding”*
>
> **A**: We have performed a comparison in Tab.1 in the supplementary to show that DCT for the local range encoder could improve the performance. We also provide a [visual comparison](https://raw.githubusercontent.com/AnonymousAut/Anonymous-Result/gh-pages/rebuttal/dct.gif). For more details about the usage of DCT, please refer to our answer on “**Usage of DCT**” in the general response.
>
> ---
>
> **Q**: *“the combined prediction of absolute joint locations vs. center of mass trajectory + relative joint locations”*
>
> **A**: For predicting “center of trajectory+relative joint locations”, the computation of the relative joint locations requires joint angle information to indicate the position of the child node to the parent node. However, this information is not available in most multi-person datasets. Most datasets only provide GT absolute joint locations since the GT is generated with the help of pose estimation techniques. Thus our method on prediction of absolute joint locations is more applicable and can be adopted and evaluated in most datasets.
>
> ---
>
> **Q**: *“The description of the discriminator is a little bit vague. One question not answered is why it is not designed as a similar architecture as the prediction part. In experiments, it is only giving marginal improvements on most of the datasets (Table 1/2/3). So this seems either a missing opportunity or deadweight in the design. The authors please clarify.”*
>
> **A**: Since the pose classification task the discriminator performs is a relatively simple task, we find using the current network can already show obvious improvement. While using a transformer for discriminator is certainly a choice, using our current architecture largely reduces the computation cost. We provide a qualitative comparison on our method with and without the discriminator and provide it [here](https://raw.githubusercontent.com/AnonymousAut/Anonymous-Result/gh-pages/rebuttal/disc.gif). We observe that with the discriminator, our method could generate a much smoother motion sequence. Especially for the single walking human in the video, the walking motion gradually diminishes when the discriminator is not used.
>
> ---
>
> **Q**: *“The MPJPE metric has no unit. Is it in mm, cm, m, or any other metric system?”*
>
> **A**: We report MPJPE in 0.1m as introduced in Ln. 248 in our paper.
>
> ---
>
> **Q**: *“Predicting future motion is by its natural non-deterministic. Here we are comparing the prediction with the "groundtruth", which is one possible instantiation among many possible futures. Is there any consideration in this aspect?”*
>
> **A**: Please refer to our answer on “**Multiple possibility prediction**” in the general response.
>
> ---
>
> **Q**: *“I didn't see much discussion about the limitation of the work.”*
>
> **A**: We believe that a potential limitation is our setting is not suitable for predicting multiple possibilities and our method is not designed for it. This could be our future work.

---

> > ### Author Response · Authors · 2021-09-02
> > **Please let us know whether you have additional questions**
> >
> > Dear Reviewer,
> >
> > We have provided more results and explanations based on your review. Since today is the last day for discussion, can you please go over them and let us know whether you have additional questions or not?
> >
> > Thank you very much.

---

> > ### Comment · Reviewer_MkB1 · 2021-09-02
> > **Updating recommendation**
> >
> > I would like to thank the authors for the detailed rebuttal. The rebuttal has addressed most of my concerns. Given that other reviewers have also provided positive feedback for this work. I am updating my rating to 6: Marginally above the acceptance threshold.
> >
> > If the paper is accepted, please kindly consider the following concerns for revision:
> >
> > - The description of the use of DCT needs justification. The reason like "other paper uses it so it is good" is not convincing.
> > - The use of the discriminator needs investigation. Only providing qualitative comparison seems insufficient. If the authors believe the unique design leads to smoother prediction, then please use quantitative measures to assess that. Otherwise, it would still be a dead weight in the method.

---

### Author Response · Authors · 2021-08-10
**General response**

Dear reviewers, we appreciate all your feedback and in the following, we provide our response to general comments and further explain the insights of our method. We welcome further discussion with each of the reviewers to address any remaining concerns.

**Why local-range and global-range**

We thank **Reviewer BvU8** for acknowledging: *“The idea of combining global and local in the proposed way is interesting. Not all global context might be important for all pose sequences, the proposed approach helps tackle that.”* and **Reviewer MkB1** on originality: *“As far as I know this method is new for the tasks of multiple person motion prediction.”*  In our framework, the local-range encoder helps to produce a smooth motion and the global-range encoder models the interaction of all the persons in the scene. We disentangle these two functions using two encoders to add our understanding of the problem as an inductive bias to the model, which facilitates the optimization.

Specifically, for local-range encoder, the task of synthesizing a natural motion based on previous states itself is actually a challenging task. To ensure the smoothness of the motion, the model requires **dense sampling** of the input sequence. For global-range encoder, we consider the interaction of all the persons in the whole scene. To help the model to focus on the interaction instead of the motion, **sparse sampling** of the sequences are used as inputs. Of course, with infinite compute bandwidth and data, one big Transformer could be enough to encode everything. However, realistically, with limited compute and especially restriction from the scale of dataset, we find the separate processes of different aspects of data largely reduces the optimization difficulty.

To further illustrate our method’s advantage, we perform experiments on learning with only one global-range Transformer without the local-range Transformer. We visualize the results and provide it in [this link](https://raw.githubusercontent.com/AnonymousAut/Anonymous-Result/gh-pages/rebuttal/local.gif). We observe that our method is able to predict active and smooth motion in the long-term future, while learning without local-range Transformer gets stuck towards the end  (We use green color to indicate input, blue color to indicate output). There are also quantitative comparisons in Tab.2 in the supplementary material. Our method is better than w/o local-range encoder in prediction accuracy in most cases.

**Usage of DCT**

Several studies [43,44,72] have shown that DCT is beneficial for modeling human motion based on its energy compaction property[4]. Mao et al. [44] claims that *“The main motivation behind this is that, by discarding the high frequencies, the DCT can provide a more compact representation, which nicely captures the smoothness of human motion, particularly in terms of 3D coordinates.”* We have also performed a comparison in Tab.1 in the supplementary to show that DCT for the local range encoder could improve the performance. At the same time, visual quality is improved and we provide a [visual comparison](https://raw.githubusercontent.com/AnonymousAut/Anonymous-Result/gh-pages/rebuttal/dct.gif). We observe without using DCT, the predicted poses drift away towards the end. (We use green color to indicate input, blue color to indicate output).

**Multiple possibility prediction**

Unlike single person motion prediction, multiple-person motion is restricted by the reactions to the other persons. Thus the uncertainty for prediction is relatively smaller here. Besides, we are predicting 1 to 3 seconds of future by taking 1 second inputs, thus we can apply a deterministic model to perform the task. Besides comparing with ground truth, we have conducted a user study to compare the naturalness of the prediction results in Tab. 4. The qualitative results are also provided in the supplementary video and website.

We do agree that diverse motion prediction [69, 72] is an interesting direction. We see our work as the first step to deal with long-term, multiple person motion prediction using Transformers. We will look into incorporating this aspect as a future extension of work.

---

### Decision · Program_Chairs · 2021-09-28

**Decision:**

Accept (Poster)

**Comment:**

The majority of the reviewers agreed that this is a solid paper that deserves acceptance. The negative assessment by one of the reviewers was considered overly harsh by the ACs. Authors are highly encouraged to address the key comments reported by reviewers as well as to implement all the improvements (as indicated by authors in the rebuttal) in the final camera-ready version.

**Consistency Experiment:**

NeurIPS has a long history of experimentation. In 2014, NeurIPS ran an experiment in which 10% of submissions were reviewed by two independent committees to quantify the randomness in the review process. This year, we repeated a variant of this experiment to see how the quality of the review process has changed over time.  This paper was part of the experiment and was therefore assigned to two committees (consisting of reviewers, an Area Chair, and a Senior Area Chair) that reached independent decisions.  If both committees made the same recommendation, this recommendation was followed. If a single committee recommended acceptance, the paper was accepted (with the exception of a few cases in which the other committee identified what we considered a fatal flaw, e.g., an error in a key result).

This copy’s committee reached the following decision: **Accept (Poster)**

The other committee assigned to the paper recommended **Reject**.  You can find the other set of reviews, along with any follow up discussion with the authors here:
https://openreview.net/forum?id=rrf6XgIS_Ek